# Synthesis and Performance of Double-Chain Quaternary Ammonium Salt Glucosamide Surfactants

**DOI:** 10.3390/molecules27072149

**Published:** 2022-03-26

**Authors:** Lifei Zhi, Xiufang Shi, Erzhuang Zhang, Chuangji Gao, Haocheng Gai, Hui Wang, Zhenmin Liu, Tieming Zhang

**Affiliations:** College of Chemistry and Biological Engineering, Taiyuan University of Science and Technology, Taiyuan 030024, China; sxf769593427@163.com (X.S.); s20202111007@stu.tyust.edu.cn (E.Z.); 201921020119@stu.tyust.edu.cn (C.G.); 201921070122@stu.tyust.edu.cn (H.G.); gxx125@126.com (H.W.); zhmliu@tyust.edu.cn (Z.L.); 2011061@tyust.edu.cn (T.Z.)

**Keywords:** D (+)-glucose δ-lactone, double-chain, surfactant, surface tension, toxicity

## Abstract

A series of double-chain quaternary ammonium salt surfactants *N*-[*N*′[3-(gluconamide)] propyl-*N*′-alkyl]propyl-*N*,*N*-dimethyl-*N*-alkyl ammonium bromide (C_n_DDGPB, where n represents a hydrocarbon chain length of 8, 10, 12, 14 and 16) were successfully synthesized from D (+)-glucose δ-lactone, *N*,*N*-dimethyldipropylenetriamine, and bromoalkane using a two-step method consisting of a proamine-ester reaction and postquaternization. Their surface activity, adsorption, and aggregation behavior in aqueous solution were investigated via measurements of dynamic/static surface tension, contact angle, dynamic light scattering, and transmission electron microscopy. An analysis of their application performance in terms of wettability, emulsifying properties, toxicity, and antibacterial properties was conducted. The results show that with increasing the carbon chain length of the C_n_DDGPB surfactants, their critical micelle concentration (CMC) increased and the pC_20_ and efficiency in the interface adsorption of the target product gradually decreased. Moreover, the influence of the hydrophobic carbon chain length on the surface of polytetrafluoroethylene (PTFE) was even greater for the wetting effect, reducing the contact angle to 32° within the length range of C8–C14. The results of the contact angle change and the wettability experiments proved that C_10_DDGPB exhibited the best wettability. The liquid paraffin and soybean oil emulsification ability of C_n_DDGPB showed an upward trend followed by a downward trend with the growth of the carbon chain, with C_12_DDGPB exhibiting the best emulsification performance. The D_long_/D_short_ ratio was far lower than 1, which indicates mixed-kinetic adsorption. The surfactants formed spherical micelles and showed a unique aggregation behavior in aqueous solution, which showed an increase–decrease–increase trend with the change in concentration. A cell toxicity and acute oral toxicity experiment showed that the C_n_DDGPB surfactants were less toxic than the commonly used surfactant dodecyldimethylbenzyl ammonium chloride (1227). In addition, at a concentration of 150 ppm, C_n_DDGPB exhibited the same bacteriostatic effect as 1227 at a concentration of 100 ppm. The results demonstrate that sugar-based amide cationic surfactants are promising as environmentally friendly disinfection products.

## 1. Introduction

Surfactants are indispensable chemicals in daily life and industrial production that can significantly reduce the interfacial tension of a liquid material due to their special structure, comprising a polar head and a nonpolar tail, which provides them with both lipophilic and hydrophilic characteristics [1]. As a result of these unique features, surfactants exhibit a variety of functions such as wetting, emulsification, softening, solubilization, foaming/defoaming, rheological changes, decontamination, and surface adjustment.

Quaternary ammonium salt cationic surfactants generally have the functions of sterilization and bacteriostasis and are often used as disinfectants and fungicides. Moreover, they can be easily adsorbed on solid surfaces, providing them with surface modification properties and antistatic, softening, and hydrophobic effects. Therefore, cationic surfactants play an increasingly important role and are widely used in a variety of fields including home textiles, personal care products, crude oil mining, material synthesis, and asphalt fuel emulsification [2,3,4,5,6]. Especially after the outbreak of the new “coronavirus epidemic” in early 2020, disinfection and sterilization using quaternary ammonium salts has played a key role in ensuring health and safety. Therefore, the production and market of quaternary ammonium salt show a trend of explosive growth, and the development of new products has also increased rapidly in the past two years. However, the common quaternary ammonium salt products are prepared from petroleum, and are characterised by great toxicity and irritation. Therefore, it is urgently necessary to develop biological quaternary ammonium salt surfactants with low toxicity that are sustainable and environmentally safe [7,8].

With the increasing concern over environmental protection, continuous efforts are being devoted to the development of “green” surfactants from production to application. Compared with traditional surfactants, sugar-based surfactants exhibit low toxicity, low irritant action, and good biocompatibility, because carbohydrates are “green” products with good biodegradability, a low environmental impact, and are made from renewable raw materials. Therefore, the research and development of sugar-based surfactants has gradually become a hot spot in the industry [9,10]. Adding sugar groups into the molecular structure of quaternary ammonium salt surfactants may improve their potential for irritation, toxicity, biodegradability, water solubility, compatibility and so on. The quaternary ammonium salt surfactants developed from sugar groups have many advantages, such as green environmental protection, natural and renewable raw materials, easy biodegradation, multi-functionality, being safe and mild to the human body, and so on [11]. Double-chain quaternary ammonium salt surfactants have two carbon chains, which provide them with enhanced attraction and penetration properties. Thus, they can penetrate into organic matter and have super bactericidal ability at a low concentration against all kinds of pathogenic microorganisms, killing bacteria, viruses, and fungi quickly, permanently, and efficiently [12]. Compared with their single-chain counterparts, double-chain quaternary ammonium salt surfactants exhibit better ability to form micelles and to reduce surface tension, and have increased water solubility and very good stability.

In this paper, a series of double-chain products containing amide bonds were successfully designed and synthesized using a two-step reaction comprising an amine-ester reaction followed by quaternization with D (+)-glucose δ-lactone, *N*,*N*-dimethyldipropylenetriamine, and bromoalkane as raw materials. This reaction route has the advantages of mild conditions, a definite reaction position, high selectivity, and stable product quality. This technological route has the prospect of industrialization. The synthesis route of the *N*-[*N*′[3-(gluconamide)] propyl-*N*′-alkyl]propyl-*N*,*N*-dimethyl-*N*-alkyl ammonium bromide (C_n_DDGPB) surfactant is shown in Figure 1. The physicochemical properties of the final product, which combines the biocompatibility advantages of the sugar group and the amide bond, are discussed in light of their theoretical significance and practical application value for the development of new nontoxic bactericidal multifunctional surfactant products.

## 2. Results and Discussion

### 2.1. Structure Identification

The chemical structures of the raw materials, intermediates, and target compounds were identified by infrared spectroscopy (IR) and proton nuclear magnetic resonance spectroscopy (^1^H-NMR, ^13^C-NMR). IR, ^1^H-NMR and ^13^C-NMR spectra of compounds are shown in the Appendix A.

The C=O stretch in D (+)-glucose-lactone is indicated by a high absorption peak at 1726 cm^−1^. The stretching vibrations of –CH_3_ and –CH_2_ are represented by the two absorption peaks at 2934 cm^−1^ and 2915 cm^−1^, respectively. The shear vibration of –NH_2_ in *N*,*N*-dimethyldipropylenetriamine correlates to the absorption peak at 1472 cm^−1^. At 1645 cm^−1^ and 1540 cm^−1^, two strong absorption peaks respectively correspond to the C=O bond stretching-vibration and the N–H bending-vibration in amide, indicating the formation of the amide linkage.

(DDGPD) Yield: 90.12% (White solid). m.p. 108.6~109.1 °C. ^1^H-NMR(DMSO, ppm):δ:1.50~1.66 (m, 4H, CH_2_CH_2_CH_2_, CH_2_CH_2_CH_2_), 2.02~2.13 (m, 6H, NCH_3_, NCH_3_), 2.16~2.22 (t, 4H, CH_2_CH_2_N, NHCH_2_CH_2_), 2.36~2.50 (m, 4H, NHCH_2_CH_2_, CH_2_CH_2_NH), 2.59~2.61 (s, 1H, CH_2_NHCH_2_), 3.10~3.14 (m, 2H, CHCH_2_OH), 3.32~3.41 (t, 1H, CHOH), 3.45~3.49 (t, 2H, CHOH, CHOH), 3.53~3.57 (m, 1H, CH_2_CHOH), 3.82~3.97 (m, 5H, OH groups from sugar part), 7.75~7.77 (t, 1H, CONH).

^13^C-NMR (DMSO, ppm):δ:27.82 (CH_2_CH_2_), 29.52 (CH_2_CH_2_), 37.14 (CH_2_CH_2_), 45.70 (CH_3_), 47.38 (CH_2_CH_2_), 48.01 (CH_2_NH), 57.86 (CH_2_N), 63.83 (CH_2_OH), 70.58 (CHOH), 71.94 (CHOH), 72.80 (CHOH), 74.08 (CHOH), 172.87 (CONH).

(C_16_DDGPB) Yield: 78.56% (Light yellow solid). m.p. 114.6~115.3 °C. ^1^H-NMR (DMSO, ppm):δ:0.82~0.85 (m, 6H, CH_3_(CH_2_)_5_, CH_3_(CH_2_)_5_), 1.03~1.06 (m, 2H, CH_2_CH_3_), 1.16~1.25 (m, 50H, (CH_2_)_12_CH_3_, (CH_2_)_13_CH_3_), 1.63~1.68 (t, 4H, CH_2_(CH_2_)_5_CH_3_, CH_2_(CH_2_)_5_CH_3_), 1.77~1.83 (m, 4H, N^+^CH_2_CH_2_, N^+^CH_2_CH_2_), 2.01~2.08 (m, 2H, CH_2_CH_2_CH_2_), 2.59~2.64 (m, 2H, NCH_2_CH_2_), 2.95~2.99 (t, 2H, CH_2_CH_2_N^+^), 3.01~3.08 (m, 6H, CH_3_N^+^, CH_3_N^+^), 3.28~3.34 (m, 6H, CH_2_CH_2_CH_2_, CH_2_CH_2_N, NHCH_2_CH_2_), 3.39~3.45 (m, 3H, CH_2_CHOH, CHCH_2_OH), 3.52~3.57 (t, 1H, CHOH), 3.91~3.94 (t, 1H, CHOH), 4.00~4.05 (t, 1H, COCHOH), 4.48~5.52 (m, 5H, OH groups from sugar part), 7.96~7.98 (t, 1H, CONH).

^13^C-NMR (DMSO, ppm):δ:14.33 (CH_3_, CH_3_), 18.96 (CH_2_CH_2_, CH_2_CH_2_), 19.60 (CH_2_CH_2_), 22.59 ((CH_2_)_13_, (CH_2_)_13_), 26.30 (CH_2_CH_2_), 29.54 (NHCH_2_), 31.81 (CH_2_CH_2_), 44.27 (CH_2_N), 50.52 (CH_3_N^+^, CH_3_N^+^), 53.09 (CH_2_N), 56.46 (N^+^CH_2_), 60.46 (CH_2_N^+^), 63.65 (CH_2_OH), 70.56 (CHOH), 71.83 (CHOH), 72.61 (CHOH), 74.05 (CHOH), 173.86 (CONH).

(C_14_DDGPB) Yield: 89.67% (Light yellow solid). m.p. 102.5~103.3 °C. ^1^H-NMR (DMSO, ppm):δ:0.82~0.85 (m, 6H, CH_3_(CH_2_)_5_, CH_3_(CH_2_)_5_), 1.03~1.06 (m, 2H, CH_2_CH_3_), 1.19~1.25 (m, 42H, (CH_2_)_10_CH_3_, (CH_2_)_11_CH_3_), 1.62~1.67 (t, 4H, CH_2_(CH_2_)_5_CH_3_, CH_2_(CH_2_)_5_CH_3_), 1.78~1.82 (m, 4H, N^+^CH_2_CH_2_, N^+^CH_2_CH_2_), 2.04~2.09 (m, 2H, CH_2_CH_2_CH_2_), 2.58~2.65 (m, 2H, NCH_2_CH_2_), 2.94~2.98 (t, 2H, CH_2_CH_2_N^+^), 3.02~3.07 (m, 6H, CH_3_N^+^, CH_3_N^+^), 3.24~3.35 (m, 6H, CH_2_CH_2_CH_2_, CH_2_CH_2_N, NHCH_2_CH_2_), 3.41~3.46 (m, 3H, CH_2_CHOH, CHCH_2_OH), 3.59~3.63 (t, 1H, CHOH), 3.90~3.95 (t, 1H, CHOH), 4.01~4.06 (t, 1H, COCHOH), 4.50~5.52 (m, 5H, OH groups from sugar part), 7.95~7.98 (t, 1H, CONH).

^13^C-NMR (DMSO, ppm):δ:14.40 (CH_3_, CH_3_), 18.94 (CH_2_CH_2_, CH_2_CH_2_), 19.95 (CH_2_CH_2_), 22.54 ((CH_2_)_11_, (CH_2_)_11_), 26.90 (CH_2_CH_2_), 28.96 (NHCH_2_), 31.74 (CH_2_CH_2_), 44.89 (CH_2_N), 50.56 (CH_3_N^+^, CH_3_N^+^), 53.07 (CH_2_N), 56.46 (N^+^CH_2_), 60.83 (CH_2_N^+^), 63.58 (CH_2_OH), 70.54 (CHOH), 71.79 (CHOH), 72.58 (CHOH), 74.02 (CHOH), 173.83 (CONH).

(C_12_DDGPB) Yield: 80.45% (Light yellow solid). m.p. 74.7~76.3 °C. ^1^H-NMR (DMSO, ppm):δ:0.84~0.87 (m, 6H, CH_3_(CH_2_)_5_, CH_3_(CH_2_)_5_), 1.04~1.07 (m, 2H, CH_2_CH_3_), 1.22~1.29 (m, 34H, (CH_2_)_8_CH_3_, (CH_2_)_9_CH_3_), 1.65~1.70 (t, 4H, CH_2_(CH_2_)_5_CH_3_, CH_2_(CH_2_)_5_CH_3_), 1.78~1.81 (m, 4H, N^+^CH_2_CH_2_, N^+^CH_2_CH_2_), 2.05~2.09 (m, 2H, CH_2_CH_2_CH_2_), 2.66~2.70 (m, 2H, NCH_2_CH_2_), 2.97~3.01 (t, 2H, CH_2_CH_2_N^+^), 3.05~3.08 (m, 6H, CH_3_N^+^, CH_3_N^+^), 3.28~3.34 (m, 6H, CH_2_CH_2_CH_2_, CH_2_CH_2_N, NHCH_2_CH_2_), 3.51~3.54 (m, 3H, CH_2_CHOH, CHCH_2_OH), 3.56~3.58 (t, 1H, CHOH), 3.91~3.93 (t, 1H, CHOH), 4.04~4.06 (t, 1H, COCHOH), 4.49~5.51 (m, 5H, OH groups from sugar part), 7.97~7.99 (t, 1H, CONH).

^13^C-NMR (DMSO, ppm):δ:14.39 (CH_3_, CH_3_), 18.98 (CH_2_CH_2_, CH_2_CH_2_), 19.58 (CH_2_CH_2_), 22.58 ((CH_2_)_9_, (CH_2_)_9_), 26.30 (CH_2_CH_2_), 29.49 (NHCH_2_), 31.80 (CH_2_CH_2_), 44.25 (CH_2_N), 50.54 (CH_3_N^+^, CH_3_N^+^), 52.75 (CH_2_N), 56.47 (N^+^CH_2_), 60.43 (CH_2_N^+^), 63.66 (CH_2_OH), 70.59 (CHOH), 71.85 (CHOH), 72.72 (CHOH), 74.12 (CHOH), 173.90 (CONH).

(C_10_DDGPB) Yield: 92.24% (Light yellow solid). m.p. 49.8~50.4 °C. ^1^H-NMR (DMSO, ppm):δ:0.81~0.83 (m, 6H, CH_3_(CH_2_)_5_, CH_3_(CH_2_)_5_), 1.18~1.24 (m, 28H, (CH_2_)_7_CH_3_, (CH_2_)_7_CH_3_), 1.61~1.64 (m, 4H, CH_2_(CH_2_)_5_CH_3_, CH_2_(CH_2_)_5_CH_3_), 1.72~1.77 (t, 4H, N^+^CH_2_CH_2_, N^+^CH_2_CH_2_), 2.27~2.30 (m, 2H, CH_2_CH_2_CH_2_), 2.36~2.40 (m, 4H, NCH_2_CH_2_, CH_2_CH_2_N^+^), 2.87~2.89 (m, 4H, CH_2_CH_2_CH_2_, CH_2_CH_2_N), 3.00~3.05 (m, 6H, CH_3_N+, CH_3_N^+^), 3.22~3.27 (m, 3H, CH_2_CHOH, CHCH_2_OH), 3.37~3.39 (m, 2H, NHCH_2_CH_2_), 3.47~3.49 (m, 2H, CHOH, CHOH), 3.54~3.56 (t, 1H, COCHOH), 3.77~4.02 (m, 5H, OH groups from sugar part), 7.91~7.94 (t, 1H, CONH).

^13^C-NMR (DMSO, ppm):δ:14.37 (CH_3_, CH_3_), 18.93 (CH_2_CH_2_, CH_2_CH_2_), 19.60 (CH_2_CH_2_), 22.57 ((CH_2_)_7_, (CH_2_)_7_), 26.27 (CH_2_CH_2_), 29.39 (NHCH_2_), 31.77 (CH_2_CH_2_), 44.28 (CH_2_N), 50.71 (CH_3_N^+^, CH_3_N^+^), 53.08 (CH_2_N), 56.45 (N^+^CH_2_), 60.42 (CH_2_N^+^), 63.59 (CH_2_OH), 70.54 (CHOH), 71.79 (CHOH), 72.70 (CHOH), 74.06 (CHOH), 173.83 (CONH).

(C_8_DDGPB) Yield: 69.18% (Light yellow solid). m.p. 47.6~48.4 °C. ^1^H-NMR (DMSO, ppm):δ:0.83~0.86 (m, 6H, CH_3_(CH_2_)_5_, CH_3_(CH_2_)_5_), 1.20~1.28 (m, 20H, (CH_2_)_5_CH_3_, (CH_2_)_5_CH_3_), 1.64~1.67 (m, 4H, CH_2_(CH_2_)_5_CH_3_, CH_2_(CH_2_)_5_CH_3_), 1.73~1.77 (t, 4H, N^+^CH_2_CH_2_, N^+^CH_2_CH_2_), 2.17~2.19 (m, 2H, CH_2_CH_2_CH_2_), 2.33~2.36 (m, 4H, NCH_2_CH_2_, CH_2_CH_2_N^+^), 2.72~2.75 (m, 2H, CH_2_CH_2_CH_2_), 2.84~2.87 (t, 2H, CH_2_CH_2_N), 3.22~2.29 (m, 6H, CH_3_N+, CH_3_N+), 3.36~3.39 (m, 3H, CH_2_CHOH, CHCH_2_OH), 3.46~3.49 (m, 2H, NHCH_2_CH_2_), 3.54~3.56 (m, 2H, CHOH, CHOH), 3.89~3.91 (t, 1H, COCHOH), 3.92~4.22 (m, 5H, OH groups from sugar part), 7.86~7.89 (t, 1H, CONH).

^13^C-NMR (DMSO, ppm):δ:14.40 (CH_3_, CH_3_), 18.94 (CH_2_CH_2_, CH_2_CH_2_), 19.61 (CH_2_CH_2_), 22.52 ((CH_2_)_5_, (CH_2_)_5_), 26.25 (CH_2_CH_2_), 29.00 (NHCH_2_), 31.64 (CH_2_CH_2_), 44.27 (CH_2_N), 50.56 (CH_3_N^+^, CH_3_N^+^), 53.07 (CH_2_N), 56.46 (N^+^CH_2_), 60.41 (CH_2_N^+^), 63.58 (CH_2_OH), 70.54 (CHOH), 71.79 (CHOH), 72.58 (CHOH), 74.02 (CHOH), 173.83 (CONH).

### 2.2. Surface Tension

Surfactant molecules were oriented on a gas/liquid interface with the hydrophilic head group facing the solution and the hydrophobic group facing outward. When the concentration of surfactant was large enough, the liquid surface was filled with a layer of oriented surfactant molecules, and the remaining surfactant molecules gathered in the solution to form micelles. The equilibrium surface tension of the C_n_DDGPB surfactant aqueous solution was determined as depicted in Figure 1. The surface tension curve belonging to the solution of surfactant fell significantly with increasing concentration until it reached a break point at low concentration. The concentration value of the abscissa corresponding to the break point was the critical micelle concentration (CMC) value. After that, a constant value of surface tension was attained, referred to as the equilibrium surface tension (γ_cmc_). The γ_cmc_ and CMC values are shown in Table 1.

The saturated adsorption capacity at the air/water interface (Γ_max_) according to the Gibbs adsorption formula can be estimated by using the slope of Figure 1 [13]:(1)$$\Gamma_{\max}=-\frac{1}{2.303nRT}{\left(\frac{\partial \gamma }{\partial \mathrm{lg}c}\right)}_{T}$$

Then, the minimum surface area occupied by a single molecule (*A*_min_) was determined by using the saturated adsorption amount in the following manner:(2)$$A_{\min}=\frac{10^{16}}{N_{A}\Gamma_{\max}}$$
where *N_A_* is Avogadro’s number.
(3)$$pC_{20}=-\log C_{20}$$

The parameters such as surfactant efficiency and efficacy are used to assess the relative potential of surfactants to lower water surface tension. The surfactant concentration (mol/L) necessary to lower the surface tension of water by 20 dynes/cm is referred to as C_20_. The effect of the surfactant on adsorption and micellization can be expressed by CMC/C_20_. Large CMC/C_20_ values are indicative of favored adsorption of the surfactant at the interface relative to micelle formation tendency. When the pC_20_ value exceeds unity, the amount of surfactant needed to create a certain surface tension reduction value is reduced by 10 times.

It can be seen from Table 1 that the cmc of the double-chain single-head surfactants C_8_DDGPB to C_14_DDGPB was 0.101–0.193 mmol/L, which is two orders of magnitude lower than that of a conventional single-chain cationic surfactant (C_n_DGPB). The cmc value increased with increasing carbon chain length. However, the cmc of common surfactants decreases as the hydrocarbon chain length increases. This may be due to the special structure of the double-chain surfactant facilitating the entanglement of the molecules. This entanglement in the molecules must be opened to form aggregates. However, this is more difficult for long-carbon chains, resulting in larger cmc values. In 1993, Menger [14] et al. reported the Gemini surfactant, which has a similar phenomenon. Compared with the single-chain glucosamine surfactant C_n_DGPB and the stellate glucosamine surfactant (C_n_DBGB), the C_n_DDGPB series exhibited lower cmc values. Upon increasing the hydrophobic carbon chain length, pC_20_ gradually decreased; the efficiency of reducing the surface tension of an aqueous solution decreases as the adsorption efficiency for the target product on the interface decreases.

Our research group [15] reported that the surface tension values of C_12_DGPB and C_14_DGPB in *N*,*N*-dimethyl-*N*-[3-(glucosaminyl)] propyl-*N*-alkyl ammonium bromide (C_n_DGPB) were 28.32 and 30.26 mN/m, respectively, which are similar to the values obtained for double-chain C_12_DDGPB and C_14_DDGPB with the same carbon chain length in the present work (29.358 mN/m and 30.897 mN/m, respectively). Meanwhile, the C_10_DDGPB surfactant had a stronger ability to reduce the surface tension of water, with a surface tension of 25.464 mN/m. In contrast, the surface tension value of C_16_DDGPB was 44.8 mN/m, which is indicative of its poor ability to reduce the surface tension of water. This result might be related to the complex molecular structure of C_16_DDGPB, in which the steric hindrance and the repulsion between the hydrophobic group and water increases, resulting in poor solubility in water.

Generally, the minimum molecular area of conventional surfactants with small hydrophilic head groups depends on the length and cross-sectional area of the hydrophobic chains [17]. Compared with the single-chain glucosamine quaternary ammonium salt surfactants C_n_DGPB and stellate glucosamine quaternary ammonium surfactants C_n_DBGB, the Γ_max_ value followed the order C_n_DGPB > C_n_DDGPB > C_n_DBGB, and the A_min_ value increased in the order C_n_DGPB < C_n_DDGPB < C_n_DBGB (the molecular interface arrangement of C_n_DGPB, C_n_DDGPB, and C_n_DBGB is shown in Figure 2). This is most likely because the single-chain surfactant molecules have a hydrophobic carbon chain, small volume, linear parallel arrangement at the water/oil interface, and dense molecular arrangement [18], and Γ_max_ increases with the hydrophobic chain length. Moreover, when the size of the hydrophobic group changes while that of the hydrophilic group remains unchanged, the value of A_min_ changes. The double-chain surfactants C_n_DDGPB with two hydrophobic carbon chains exhibit a structure similar to that of the Gemini surfactant. In the complex molecular structure, the repulsion force between the two carbon chains increases, and a single molecule occupies the minimum surface area at the interface, which is larger than that of a single-chain surfactant, resulting in a smaller saturated adsorption capacity (*A*_min_) than that of the latter. Stellate surfactant C_n_DBGB has two hydrophobic carbon chains and two hydrophilic groups centered on the N atom, forming a nearly spherical structure with large steric hindrance. This causes a large repulsion between the hydrophobic group and water and a small number of stellate surfactant molecules on the interface, resulting in a large effective area of each molecule.

The time necessary for adsorption to reach equilibrium on the solution surface was not addressed in the evaluation of surfactant properties at adsorption equilibrium upon the surface of the solution as stated above. While a balanced rate of surface tension reduction is critical for some applications, others need both balanced as well as dynamic reductions in surface tension. Surfactant efficiency in lowering equilibrium surface tension is not always related to surfactant efficiency in lowering dynamic surface tension [19]. Because the properties of the interface are modified by surfactants as they adsorb, it is more important to study the dynamic surface tension in comparison to the study of equilibrium surface tension when it comes to the practical applications of surfactants.

The microdiffusion adsorption rate of surfactant molecules is known to affect their macro-performance [20]. Therefore, the change in the surface tension of C_n_DDGPB with time was tested using the maximum bubble pressure method, and the effect of the length of the hydrophobic carbon chain on the surface tension was explored.

Generally, according to Rosen’s theory [21], surface tension plots with time can be divided into four regions: induction, rapid decline, medium equilibrium, and equilibrium. As shown in Figure 3, the induction and rapid decline regions were detected for C_n_DDGPB. At low solution concentrations, the surface tension did not reach the minimum value in equilibrium within the effective detection range; however, this value was reached at high solution concentrations. Therefore, in a certain range, the increase in the concentration of C_n_DDGPB surfactant solution facilitates the reduction of the surface tension of the solution.

Easton detailed the adsorption mechanism of surfactant molecules at low solution concentrations in his study on the dynamic adsorption model of a surfactant [22]. First, molecular diffusion takes place from the bulk phase to the surface layer, then from the very thin surface layer to the subsurface layer. Later, Fanerman quantified the model using an asymptote equation, as shown in Equation (4):(4)$$\Gamma (t)=2C_{0}\sqrt{\frac{Dt}{\Pi }}-2\sqrt{\frac{D}{\Pi }}\int_{0}^{\sqrt{t}}C_{s}d(\sqrt{t-\tau })$$

*C*_0_ refers to the bulk concentration of the surfactant solution, whereas the concentration of the surfactant within the subsurface layer is represented by *C_S_*. The diffusion coefficient of the surfactant molecule is given by *D*, and *τ* is a dummy variable. As the concentration of the lower layer increases, the initial part of the equation represents the molecular migration from the volume phase to the lower layer, whereas the molecular diffusion from the lower layer back to the volume phase is represented by the second portion. This problem was solved asymptotically by Miller et al. [23,24].

When *t* → 0, i.e., *γ*(*t*) approaches *γ*_0_, surfactant molecules do not undergo back diffusion; therefore, Equation (5) is obtained after simplification of Equation (4):(5)$$\gamma {\left(t\right)}_{t\to 0}=\gamma_{0}-2nRTC_{0}\sqrt{\frac{Dt}{\Pi }}$$

The subsurface concentration is close to the volume concentration, when *t* → ∞, i.e., *γ*(*t*) approaches *γ_eq_*, hence, the integral can be unified as *t*→ ∞ without considering the change in *C_S_*:(6)$$\gamma {\left(t\right)}_{t\to \infty }=\gamma_{eq}+\frac{nRT\Gamma_{eq}^{2}}{C_{0}}\sqrt{\frac{\Pi }{4Dt}}$$

*γ_eq_* refers to the equilibrium surface tension and *γ*(*t*) denotes the surface tension at time *t*, for 1-1 ionic surfactant (*n* = 2), and Γ*eq* refers to the equilibrium excess surface concentration (Γ*_eq_* is approximately assumed to have a value equal to equilibrium surface tension, Γ_cmc_).

In the initial stage of adsorption, molecules are generally relatively easy to adsorb on the interface, and the process is diffusion-adsorption-controlled. In contrast, in the later stage, only surfactant molecules with the same orientation and sufficient energy can be adsorbed on the interface via a barrier adsorption-controlled process. According to Equations (5) and (6), the *γ* and *t*^1/2^ and the *γ* and *t*^−1/2^ values can be obtained, respectively, as shown in Figure 4.

The diffusion coefficient of conventional alkane surfactant monomers is generally 10^−10^ m^2^/s, while the diffusion coefficient of C_n_DDGPB in Table 2 is significantly lower than the order of 10^−10^ m^2^/s, indicating that the adsorption energy of C_n_DDGPB is lower than that of conventional alkane surfactant monomers.

For conventional surfactants, the effective diffusion coefficients in the early stage of adsorption (D_short_) and the late stage of adsorption (D_long_) decrease with increasing carbon chain length [25]. The same trend was observed for the C_n_DDGPB surfactants. In addition, the ratio of D_long_/D_short_ was much lower than unity, which means that D_long_ was very different from D_short_. This could be because long-chain surfactants have more steric hindrance in the molecular diffusion at the interface and the subsequent rearrangement reaction. Besides, the difference in CMC (the number of dissolved monomers within the solution) could also significantly affect adsorption kinetics [26]. This shows that the late stage of C_n_DDGPB adsorption is a mixed diffusion–kinetic adsorption rather than diffusion-controlled adsorption.

### 2.3. Wettability Study

Wetting solid surfaces with surfactant solutions plays an important role in many industrial and daily life processes such as flotation, decontamination, oil production, paint, coating, and deposition [27,28]. Therefore, it is of high theoretical and practical value to study the wettability of a surfactant solution on a solid surface. To this purpose, polytetrafluoroethylene (PTFE), a widely applied high-molecular-weight polymeric, translucent, white wax with good resistance to cold, heat, acids, alkalis, and various organic solvents and a low-energy hydrophobic surface [29], is usually used as a typical representative of hydrophobic surfaces [30]. In the present study, the investigation of the surface properties was performed by measuring the contact angle of PTFE using the seat drop test, and the wettability and adsorption of the surfactant solution on the solid surface were evaluated according to the results.

Figure 5 shows the changes in the contact angle with the surfactant concentration when the double-chain glucosamine quaternary ammonium salt surfactants C_n_DDGPB were contacted with PTFE for 100 s. As can be seen, the contact angle on the PTFE surface decreased with increasing surfactant concentration. Figure 6 shows the droplet image of C_10_DDGPB after contacting PTFE for 100 s, which is consistent with the change trend of the contact angle. Among the series, C_8_DDGPB, C_10_DDGPB, C_12_DDGPB, and C_14_DDGPB showed the same wettability of the PTFE surface; within the investigated concentration range, the contact angle was reduced to 32°. In contrast, C_16_DDGPB exhibited the worst wettability, only reducing the contact angle to 83° within the investigated concentration range. This result demonstrates that the wetting effect of the double-chain glucosamine quaternary ammonium salt on the PTFE surface is greatly affected by the carbon chain length of the hydrophobic group.

The wettability of C_n_DDGPB with different carbon chain lengths to canvas was measured, and the results are shown in Figure 7. The wettability of sugar-based surfactants is closely related to the length of their hydrophobic and hydrophilic groups. Thus, the best wettability is achieved when the ratio of the size of the hydrophilic group and the length of the hydrophobic group is appropriate, and wettability is reduced as the length of hydrophilic or hydrophobic groups increases [31]. As can be deduced from Figure 7, C_10_DDGPB had the shortest wetting time and C_16_DDGPB had the longest wetting time. Taken together, the results of the change in contact angle and wettability experiments demonstrate that C_10_DDGPB had the best wettability among the samples evaluated.

### 2.4. Emulsifying Performance

The emulsifying ability is one of the most important parameters for evaluating the performance of surfactants [32,33]. To evaluate the emulsifying effect of the CnDDGPB surfactants on an edible oil and a mineral oil, soybean oil and liquid paraffin were selected. Soybean oil is a typical vegetable oil, and liquid paraffin is, together with kerosene, gasoline, and diesel, a representative example of mineral oils, which are a mixture of refined liquid hydrocarbons obtained from petroleum. The ability of the sample solution to emulsify soybean oil and liquid paraffin was determined using the cylinder method, which consists of recording the time required to separate 10 mL water from different emulsions. Longer times are indicative of better emulsifying properties. The results are shown in Figure 8.

According to Figure 8, the emulsifying ability of C_n_DDGPB for soybean oil was significantly stronger than that for liquid paraffin. This may be related to their molecular structure and intermolecular interaction. Specifically, the main component of liquid paraffin is an alkane, while the main component of soybean oil is a fatty acid. The structure of the latter is similar to that of the surfactant, and therefore, both compounds are more compatible. The emulsifying capacity of the C_n_DDGPB series for liquid paraffin and soybean oil first increased and then decreased with the growth of the carbon chain, with C_12_DDGPB showing the best emulsifying performance. The emulsifying performance is related to the molecular structure, the hydrophilic lipophilic balance (HLB) value, and the formation of an interfacial charge in the adsorption membrane, which provides a theoretical basis for the practical application of double-chain quaternary ammonium salts as emulsifiers in the future. The good emulsifying performance of C_n_DDGPB for soybean vegetable oil indicates that it could be applied to emulsify oil and washing products.

### 2.5. Aggregation Behavior in Aqueous Solution

Amphiphilic compounds solubilize as monomers in aqueous solutions. To avoid its hydrophobic group being a subject of repulsion, the molecule rotates in search of the energetically stable state, such that the hydrophilic moiety remains in water while the hydrophobic group extends towards the air [34]. The hydrophobic groups tend to lie close to each other as the concentration rises, reducing the contact area with water and gradually arranging the molecules on the surface of the water, creating a monolayer. Different forms of micelles, such as rod micelles, spherical micelles, vesicles, and vermicelles, can form in the solution as the concentration rises above 10 times the CMC. Transmission electron microscopy (TEM) and dynamic light scattering measurements were used to study the aggregation, shape, and size of the C_n_DDGPB surfactants in an aqueous solution.

The TEM images displayed in Figure 9 show that all the C_n_DDGPB surfactants formed spherical micelles in aqueous solution at a concentration of 5.0 mmol/L. As can be seen from Figure 10, as the concentration of the C_n_DDGPB surfactant increased, the aggregate size first increased, then decreased, and then increased again, as was previously observed for the stellate lactose amide quaternary ammonium surfactants (C_n_DBLB) [35].

Figure 11 depicts a model of micellar size change for C_n_DDGPB. From the perspective of the molecular structure of the C_n_DDGPB surfactant, at low concentration, the molecular structure was dominated by a double-chain long-carbon chain configuration and was in a fully extended state, with a small number of molecular hydrophobic groups gathered together to form small micelles. As the concentration increased, more surfactant molecules were immediately aggregated, producing larger micelles. Upon further increasing the concentration, the intermolecular electrostatic repulsion was enhanced, which resulted in the bending of the hydrophilic chain and the two hydrophobic chains, affording smaller micelles. As the concentration increased, the bent surfactant molecules bound together into larger micelles. Therefore, the size of molecular structures is an important factor affecting the formation of surfactant aggregates.

### 2.6. Analysis of Cell Viability

Surfactants have been widely employed as detergents since their discovery more than half a century ago, and a clear stimulating impact on skin was initially recognized [36]. Their potential hazard to human safety when in contact with them, as well as the environmental impact of surfactant-containing wastewater discharge into rivers, were later discovered. The majority of surfactants have some cytotoxicity. The degree of toxicity of surfactants can be ranked in the following sequence. Cationic surfactants > anionic surfactants > nonionic surfactants, according to the report by Hu et al. [36,37].

An MTT test was employed for evaluating the toxicity of C_n_DDGPB surfactants towards HeLa cells in this study. The results were then viewed and photographed after live/dead cell staining, using a laser confocal microscope to investigate the interrelation between C_n_DDGPB and cytotoxicity to furnish a theoretical and experimental foundation for the prospective use of surfactants based on glycosylamide quaternary ammonium salt.

Figure 12 shows the results of the MTT assay for HeLa cells cocultured with C_n_DDGPB (concentration range: 5 to 100 μg/mL) for 48 h. In comparison to the various control groups, the cell survival rate was found to be decreased as the concentration was increased, and the toxicity was observed to have increased.

As can be seen in Figure 12, cell survival rate decreased with an increase in carbon chain length at the same concentration. C_10_DDGPB exhibited the lowest toxicity. Dodecyldimethylbenzyl ammonium chloride (1227) is a commonly used and efficient broad-spectrum cationic surfactant. A comparison of the toxicity results of 1227 and C_n_DDGPB (Figure 12) revealed that the cell survival rate of C_n_DDGPB was much higher than that of 1227, which suggests that the double-chain glucosamine quaternary ammonium salt is safer than this conventional disinfectant in terms of disinfection and sterilization. The IC_50_ values of 1227, C_8_DDGPB, C_10_DDGPB, C_12_DDGPB, C_14_DDGPB and C_16_ DDGPB are 3.31 μg/mL, 10.75 μg/mL, 13.65 μg/mL, 9.07 μg/mL, 8.36 μg/mL and 9.15 μg/mL, respectively. The CMC values of C_8_DDGPB, C_10_DDGPB, C_12_DDGPB, C_14_DDGPB and C_16_ DDGPB are 48.68 μg/mL, 60.18 μg/mL, 76.39 μg/mL, 109.24 μg/mL, and 752.59 μg/mL, respectively. This indicates that the IC_50_ value of this kind of quaternary ammonium salt is below the CMC value.

A fluorescence microscope was used for the observation of cell death. The findings are displayed in Figure 13. While the live cells are yellow-green, the dead cells can be observed in red. At low concentrations, practically all of the cells were yellow-green, however, at higher concentrations, the number of live cells was dramatically decreased, corresponding to the cell survival rate. At low concentrations, the C_n_DDGPB surfactants are minimally-toxic or non-toxic, and their toxicity increases with a rise in concentration.

### 2.7. Acute Oral Toxicity Test Results

The toxicity of the surfactants was assessed on the basis of the toxicity evaluation of disinfectants, according to which nontoxic disinfectants have an LD_50_ larger than 5000 mg/kg body weight, low-toxic disinfectants exhibit an LD_50_ larger than 1000 mg/kg body weight, and those having an LD_50_ greater than 100 mg/kg have medium toxicity. The survival status of ICR mice fed with different concentrations was observed for 14 days. The results are summarized in Table 3, which reveals that the mortality of mice in the 1227 test group was the highest, with an LD_50_ of mice larger than 100 mg/kg, indicating that the surfactant 1227 has medium toxicity. The mortality of mice in the C_10_DDGPB test group was significantly lower than that in the C_12_DDGPB test group, indicating that the toxicity of C_10_DDGPB was lower than that of C_12_DDGPB, which is consistent with the results of the cytotoxicity experiment. No obvious blackening or body stiffness was observed in any of the tested animals during the 14-day observation period. The LD_50_ of mice was larger than 1000 mg/kg, indicating that the double-chain glucosamine quaternary ammonium surfactants of the C_n_DDGPB series have low toxicity.

### 2.8. Antibacterial Properties of the Surfactants

In the context of the recent new coronavirus epidemic and the spread of the delta mutant strain, disinfection and sterilization are key actions to ensure health and safety. Quaternary ammonium surfactants have been used as bactericides and disinfectants for nearly a century. At present, they are widely used in the household chemical, food, medicine, and healthcare industries, among other fields [38]. In aqueous solution, the quaternary ammonium ions of a quaternary ammonium salt surfactant are positively charged and can be adsorbed on the negatively charged microbial surface to form microclusters, which changes the permeability of the cell membrane and kills microorganisms. Positively charged quaternary ammonium ions can also destroy the envelope of a virus and change its structure, resulting in virus inactivation [39]. Many studies have shown that the length of the hydrophobic chain, the charge density, and the adsorption of quaternary ammonium ions have a great impact on antibacterial activity [40]. In quaternary ammonium salt bactericides, the N atom on the head group generally has a positive charge, and the ability to adsorb to the bacterial cell wall increases with the density of the positive charge [41].

The bacteriostatic rate was calculated according to Equation (7), and the results are shown in Figure 14.
(7)$$\mathrm{Bacteriostatic}\;\mathrm{rate}=\frac{\mathrm{Average}\;\mathrm{colony}\;\mathrm{count}\;\mathrm{of}\;\mathrm{control}\;\mathrm{sample}-\mathrm{average}\;\mathrm{colony}\;\mathrm{count}\;\mathrm{of}\;\mathrm{test}\;\mathrm{sample}}{\mathrm{Average}\;\mathrm{colony}\;\mathrm{count}\;\mathrm{of}\;\mathrm{control}\;\mathrm{sample}}$$

As can be gleaned from Figure 14, the antibacterial activity of C_n_DDGPB was lower than that of 1227. At a concentration of 50 ppm, the bacteriostatic rate of 1227 against Escherichia coli and Staphylococcus aureus reached 92.36% and 100%, respectively, and 100% in all cases at 100 ppm. The maximum inhibitory rate of C_n_DDGPB against Escherichia coli was 89.72% at 100 ppm and 100% at 150 ppm, which is most likely because C_n_DDGPB was synthesized from sugars and had low toxicity. Combined with the cytotoxicity and acute oral toxicity results described in Section 2.7 and Section 2.8, respectively, these results demonstrate that the toxicity of C_n_DDGPB was much smaller than that of 1227. A comprehensive analysis showed that although the bacteriostatic rate of 1227 was high at the same concentration, it was highly toxic. The same bacteriostatic effect can be achieved using C_n_DDGPB at higher concentrations, which is safer and more environmentally friendly. Overall, the results presented here reveal this sugar-based surfactant as a promising environmental protection disinfection and sterilization product.

## 3. Materials and Methods

### 3.1. Materials

D (+)-glucose δ-lactone (purity ≥ 98%, Aldrich Company); 1-Bromooctane, 1-Bromodecane, 1-Bromododecane, 1-Bromotetradecane, 1-Bromohexadecane (chemically pure, Shanghai Bohua Biochemical Reagent Co., Ltd., Shanghai, China); *N*,*N*-Dimethyldipropylenetriamine (purity ≥ 99%, Zhangjiagang Dawei Additives Co., Ltd., Shanghai, China); absolute ethanol, methanol (analytical purity, Shanghai Lingfeng Chemical Reagent Factory, Shanghai, China); liquid paraffin (chemically pure, Tianjin Kemio Chemical Reagent Co., Ltd., Tianjin, China); soybean oil (edible grade, Yihai Jiali Food Marketing Co., Ltd., Shanghai, China).

The HeLa cell line was provided by Shanghai Saibaikang Biology, Ltd., Shanghai, China.

### 3.2. Equipment

Hitachi 270-30 FT-IR spectrometer (Hitachi, Tokyo, Japan); nuclear magnetic resonance instrument (Varian Inova-400MHz, DMSO solvent); RE-52A rotary evaporator (Shanghai Yarong Biochemical Instrument Factory, Shanghai, China); ESJ200-4 Electronic Balance (Shenyang Longteng Electronics Co., Ltd., Shenyang, China).

K12 surface tension meter (Krüss, Hamburg, Germany); Bp-100 dynamic surface tension meter (Krüss, Germany); DLS (nano ZS90, Malvern, UK); JEM-1011EX transmission electron microscope (TEM) (Japan Electronics Corporation); contact angle measuring instrument (JC2000C, Shanghai Zhongchen Digital Technology Equipment Co., Ltd., Shanghai, China); micro melting point instrument (JM624).

CO_2_ constant temperature incubator (WIGGENSWCI-180, Beijing Sangyi Experimental Instrument Research Institute, Beijing, China); centrifuge (TD5, Shanghai Luxiangyi Centrifuge Instrument Co., Ltd., Shanghai, China); laser confocal microscope (IX73, OLYMPUS); constant temperature and humidity incubator (LHS-150HC, Shanghai Huitai Instrument Manufacturing Co., Ltd., Shanghai, China); stainless steel portable pressure steam sterilizer (Shanghai Boxun Industrial Co., Ltd., Shanghai, China).

### 3.3. Sample Synthesis

The synthetic route of double-chain quaternary ammonium salt glucosamide surfactants (C_n_DDGPB, *n* = 8, 10, 12, 14, 16) was shown in Figure 1.

Synthesis method of intermediate (DDGPD): A mixture of D (+)-glucose δ-lactone (150 mmol), *N*,*N*-dimethyldipropylenetriamine (170 mmol) and methanol (250 mL) was stirred at reflux temperature for 12 h. The solvent was removed by evaporation, the residue was washed three times with ether and dried under reduced pressure to a constant mass.

Synthesis method of C_n_DDGPB: A mixture of DDGPD (50 mmol), bromoalkane (120 mmol) and ethanol (150 mL) was stirred at reflux temperature for 36 h. The solvent was removed by evaporation, the residue was washed three times with ether and dried under reduced pressure to a constant mass.

### 3.4. Equilibrium Surface Tension Measurement

Doubly distilled water was employed to make fresh surfactant stock solutions which were subsequently diluted to the desired concentrations. Prior to the measurements, the solutions were aged for 24 h. A processor tension meter K12 (Krüss, Hamburg, Germany) was used to measure the surface tension using the ring approach. The temperature was set to 25.0 ± 0.1 °C, and the apparent surface tension measurements were taken five times, with a 2 min gap between consecutive measurements, and then the average value was taken. The turning point in the surface tension versus concentration curve was used to establish the critical micelle concentration (CMC).

### 3.5. Wettability Test

Using ultrapure water, a set of double-chain quaternary ammonium salt glucosamide surfactant solutions was made. Contact angle measuring equipment was used to measure the contact angles of the polytetrafluoroethylene film (PTFE) and the solution at 25 °C. The sessile drop method was used to determine the contact angle of droplets with the PTFE surface. A charge-coupled device (CCD) camera was employed to dynamically photograph the shape of the droplet during the spreading process, and image processing software was used to estimate the contact angle. Before each measurement, the contact angle of deionized water on PTFE was determined, and the experimental temperature was kept constant at 25.0 ± 0.2 °C.

According to the national standard GB/T 11983-2008 method for measuring the wetting power of surfactants, the canvas settlement test was adopted. In this test, the time that passed from the moment when the canvas entered the solution until it started to sink after complete wetting was recorded, and this measurement was repeated five times to obtain the average value.

### 3.6. Emulsifying Performance

The emulsifying performance [42] was determined using the measuring cylinder method. Typically, 40 mL (1 g/L) of the measured liquid and 40 mL of liquid paraffin/edible soybean oil were placed in a 500 mL iodine measuring bottle, which was shaken vigorously up and down five times and then allowed to stand for 1 min. This sequence was repeated five times. After the fifth oscillation, the liquid was immediately poured into a 100 mL measuring cylinder, and a stopwatch was started to record the time required for 10mL water phase separation. This process was repeated five times to obtain the average value.

### 3.7. Dynamic Surface Tension Measurement

By ascertaining the maximal pressure needed for the liquid to blow out a bubble from the capillary tip (referred to as the maximum bubble pressure method (MBPM)), the dynamic surface tension was determined. The effective surface time was 0.01–25 s, and the experimental temperature was 25 °C.

### 3.8. Transmission Electron Microscopy (TEM)

Copper online Formvar grids were drop casted with the surface-active solution, followed by dyeing to prepare the TEM samples. Careful contact of a filter paper with one end of the grid resulted in the removal of the extra liquid. A phosphotungstic acid aqueous solution of 2% was used after drying to keep the dyeing time within 60–90 s. Absorption of the excess solution was accomplished by running a filter paper at the edge of the copper mesh. Subsequently, the specimens were air-dried naturally for a few hours and subjected to an electron microscope analysis.

### 3.9. Dynamic Light Scattering Experiment

The incoming light source for the dynamic light scattering (DLS) apparatus was a 22 mW He–Ne laser with a scattering angle of 90° and a wavelength of 632.8 nm. The specimen was allowed to filter via a 0.45 μm pore size membrane prior to being transferred to a specialized cuvette. The temperature was kept to a constant 25.0 ± 0.1 °C. The process was repeated three times for each sample.

### 3.10. Cytotoxicity Test

In order to detect cytotoxicity, an MTT assay was employed. Test specimens of 10 mg were weighed at 2 mg/mL, followed by sterilization on an ultra-clean working table with UV irradiation for 30 min. The experiment was split into two groups: control and experimental. The control group received 100 μL of complete medium for each well. C_n_DDGPB samples were added to the experimental group at 5 μg/mL, 10 μg/mL, 25 μg/mL, 50 μg/mL, 75 μg/mL, and 100 μg/mL, repeated 3 times for each sample. HeLa cells (cervical cancer cells) in the logarithmic growth phase were harvested, and cell counts were performed. Adjustments were made for cell concentrations, and using a 96-well plate, 4 × 10^3^ cells/well were seeded followed by 48 hours’ incubation in 5% CO_2_ in a constant temperature incubator at 37 °C. The medium was removed. Individual wells were cleaned three times with PBS (phosphate buffer), then media comprising 10% MTT were added at a specific amount (100 μL/well) in the presence of 5% CO_2_, and set to incubate at 37 °C for 4 h. Following removal of the supernatant, DMSO (100 μL) was put into individual wells. After shaking for 10 min, the absorbance(A) was recorded at 570 nm. Cell viability (%) = (A_experimental group_ − A_background group_)/(A_control group_ − A_background group_) × 100%. GraphPad Prism 5 software was used to calculate the value of 50% inhibition concentration (IC_50_).

The cells were incubated in varying concentrations by employing the live/dead kit (Calcein-AM/PI double staining). The dead cells were yellow-green and red whereas the healthy cells were excited at 490 ± 10 nm in a fluorescence microscope.

### 3.11. Acute Oral Toxicity Test

The acute oral toxicity test was conducted according to the determination method described in Part II of disinfection technical specifications (2002 Edition). Briefly, each sample was divided into three portions with concentrations of 0.15, 0.03 and 0.003 g/mL, respectively, which were administered to ICR mice by gavage three times a day in amounts of 0.2, 0.2 and 0.3 mL. Then, the mice were observed for 14 days. (The experiment was completed by Guangdong Detection Center of Microbiology, and met the general requirements for the ability of testing and calibration laboratories (ISO/IEC 17025:2017).)

### 3.12. Antimicrobial Activity

The antibacterial experiment was operated according to the evaluation method of the antibacterial effect of daily chemical products described in the national standard GB/T 2738-2012 as follows: The test sample was diluted with sterile distilled water to the required concentration (100 or 50 ppm). The fresh culture corresponding to the 3rd–14th generation (cultured on a nutrient agar medium for 18–24 h) of the strain was taken and properly diluted with a PBS solution to prepare a bacterial suspension (specifically, 0.1 mL of sample was dropped into 5.0 mL of PBS control solution to achieve a number of recovered bacteria of 1 × 10^4^–9 × 10^4^ cfu/mL). Then, 5.0 mL of the test sample stock solution or its diluent was placed into a sterilization test tube and maintained at 20 °C for 5 min. Subsequently, 0.1 mL of test bacterial solution was added into the test tube and mixed quickly. After 5 min, 0.5 mL of the mixture of test bacteria and sample was taken, added to a sterilized test tube containing 4.5 mL of PBS, and mixed. After 10 min, 0.5 mL of the sample solution (or the diluent with two dilutions after appropriate dilution) was placed in a sterilization plate. Two sterilization plates were inoculated with the sample solution or the dilution, and 15 mL of nutrient agar medium was poured at 40 °C–45 °C. The plates were rotated to obtain a uniform mixture, turned after the agar solidified, and cultured at 35 °C for 48 h. The viable bacterial colonies were then counted.

To prepare a control sample, the test sample was replaced with PBS and the abovementioned steps were followed. The experiment was repeated three times to obtain the average value.

## 4. Conclusions

A series of double-chain quaternary ammonium salt surfactants (C_n_DDGPB, where n represents a hydrocarbon chain length of 8, 10, 12, 14 and 16) were successfully synthesized from D (+)-glucose δ-lactone, *N*,*N*-dimethyldipropylenetriamine, and bromoalkane using a two-step method consisting of a proamine-ester reaction and postquaternization. Their surface activity, adsorption, and aggregation behavior in aqueous solution were investigated via measurements of dynamic/static surface tension, contact angle, dynamic light scattering, and transmission electron microscopy. An analysis of their application performance in terms of wettability, emulsifying properties, toxicity, and antibacterial properties was conducted. The results show that, with increasing the carbon chain length of the C_n_DDGPB surfactants, their cmc increased and the pC_20_ and efficiency in the interface adsorption of the target product gradually decreased. Compared with the single-chain glucosamine quaternary ammonium salt surfactants C_n_DGPB and the stellate glucosamine quaternary ammonium surfactants C_n_DBGB, the Γ_max_ value followed the order C_n_DGPB > C_n_DDGPB > C_n_DBGB, and the A_min_ value increased in the order C_n_DGPB < C_n_DDGPB < C_n_DBGB. Moreover, the influence of hydrophobic carbon chain length on the surface of polytetrafluoroethylene (PTFE) was even greater for the wetting effect, reducing the contact angle to 32° within the length range of C8–C14. The results of the contact angle change and the wettability experiments proved that C_10_DDGPB exhibited the best wettability. The liquid paraffin and soybean oil emulsification ability of C_n_DDGPB showed an upward trend followed by a downward trend with the growth of the carbon chain, with C_12_DDGPB exhibiting the best emulsification performance. The D_long_/D_short_ ratio was far lower than 1, which indicates mixed-kinetic adsorption. The surfactants formed spherical micelles and showed a unique aggregation behavior in aqueous solution, which showed an increasedecrease–increase trend with the change in concentration. A cell toxicity and acute oral toxicity experiment showed that the C_n_DDGPB surfactants were less toxic than the commonly used surfactant dodecyldimethylbenzyl ammonium chloride (1227). In addition, at a concentration of 150 ppm, C_n_DDGPB exhibited the same bacteriostatic effect as 1227 at a concentration of 100 ppm. The results demonstrate that sugar-based amide cationic surfactants are promising as environmentally friendly disinfection products.

## Data Availability

Not applicable.

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
