# Peer review of "Synthesis and Performance of Double-Chain Quaternary Ammonium Salt Glucosamide Surfactants"

_molecules, 2022, doi:10.3390/molecules27072149_

Round 1

Reviewer 1 Report

The present manuscript deals with the synthesis, chemical physical characterization and biological evaluation of double-chain quaternary ammonium salt glucosamide surfactants. The experimental study conducted is comprehensive and adequate. Some comments are:

Introduction: The rationale about the proposed surfactants could be better highlighted

A summary table and overview for chemical structures about the synthesized surfactants could be useful.

Synthesis procedure is not reported in the text with the exception of scheme synthesis in Figure 19.

 It is not reported how the CMC values were calculated,

The method used for the emulsifying performance test is not clear (Paragraph 3.6).

Why the CMC values of the synthesized surfactants increase with the elongation of the hydrocarbon chain? Generally, CMC decreases for surfactants with a longer hydrocarbon chain.

“Emulsifying time” is not a correct expression in the caption of Figure 10. Please improve the caption.

IC50 values from data in Figure 16 could be calculated and compared with the CMC values for surfactants.

Reviewer 2 Report

see attached file

Round 2

Reviewer 1 Report

The authors have answered to my comments and the manuscript is suitable for publication

Author Response

Thank you.

Reviewer 2 Report

References are significantly improved and now accessible to a much broader readership!

It appears that the authors have adequately addressed most of the comments. However, the manuscript is still missing measurement uncertainties for most of the reported data. While some data does include estimated uncertainty, see Figure 16 and Table 3, much of the data does not express associated uncertainty. Readers cannot assess significance without uncertainties. As one example, in Table 1, Amin reports area values that vary from 3 to 7 significant figures. How can this one measure vary from ~1 part in 100 for some surfactants to ~1 part in 1,000,000 for others?

Granted, it is important that replicate analyses were performed and averages reported (as should be always be done), but the standard deviation or relative uncertainty or some measure of uncertainty needs to accompany all of the reported data, whether that be included in the figures or stated in tables or the experimental section.

Once this is corrected, the manuscript will be publishable.

Author Response

Ms. Ref. No.: Molecules-1577205

Title: Synthesis and Performance of Double-chain Quaternary Ammonium Salt Glucosamide Surfactants

Dear Editor:
   Modified place in the manuscript has been dyed red. Feedback comments are appended below.

Response to Reviewer 2 Comments:

Point 1: It appears that the authors have adequately addressed most of the comments. However, the manuscript is still missing measurement uncertainties for most of the reported data. While some data does include estimated uncertainty, see Figure 16 and Table 3, much of the data does not express associated uncertainty. Readers cannot assess significance without uncertainties. As one example, in Table 1, Amin reports area values that vary from 3 to 7 significant figures. How can this one measure vary from ~1 part in 100 for some surfactants to ~1 part in 1,000,000 for others?

Amin was calculated, not measured. We have modified and unified the significant figures of the data in Table 1. Other data were also carefully examined.

English language and style has been carefully checked and modified.

The manuscript has been carefully revised,If there is any problem about this manuscript, please let us know. Thank you.

 Sincerely

 Lifei Zhi